# Exploring the Rational and Supernatural: Wang Chong's Critical Analysis of Ghosts and Deities in Han Dynasty Customs

**Xiaofei Ma**

Department of History, Qingdao University, Qingdao 266071, China; maxiaofei6@126.com

**Abstract:** This paper examines the critical perspectives of scholars during the Han Dynasty on customs and beliefs related to ghosts and deities. Focusing on Wang Chong as an example, it explores the naturalistic explanations of life and death, the concept of ghosts and deities, and the associated customs of funerals, sacrifices, and taboos. Wang Chong's criticisms focused on the core ideologies that underpinned funeral practices, sacrifices, and taboos and attempted to undermine the essence of these traditional customs. By reinterpreting funeral practices, sacrifices, and taboos from a ritualistic perspective that emphasized the social function rather than their supernatural implications, Wang Chong aimed to reconcile local tradition with rationality and promote a more profound understanding of the world. His approach, though complex and at times seemingly contradictory, holds an important position among the intellectual critiques of customs and beliefs during the Han Dynasty, and it sheds light on the challenges faced by ancient Chinese scholars in navigating the intersections between rationality, morality, and religion.

**Keywords:** Wang Chong; the Han Dynasty; ghosts and deities; customs; rationality

## 1. Introduction

The Han Dynasty was an era that attached great importance to customs. The term "customs" refers to the collective habits and practices exhibited by a society across various aspects such as politics, economy, culture, and ethical attitudes (Gong 2005, p. 41). Scholars in the Han Dynasty generally believed that customs were crucial indicators of political conditions, with well-established customs reflecting successful governance. Consequently, they paid attention to collecting customs from different regions across the country to observe political successes and losses. At the same time, they actively employed administrative orders and used cultural approaches to reshape diverse customs, aiming to guide them along the path they deemed appropriate. Historical records abound with instances of magistrates' efforts to transform local customs, while some scholars upheld the view that "When the *ru* are in the court, they improve the government. When they are in subordinate positions, then they improve the state's customs [儒者在本朝则美政，在下位则美俗]" (Hutton 2014, p. 54), contributing to the transformation of customs through criticism.

Regarding the customs of the Han Dynasty, the belief in ghosts and deities, which viewed non-human entities such as "ghosts" and "deities" as sources of extra-human power, was closely intertwined with people's daily lives. The concept of ghosts and deities, along with related practices such as funeral rites, sacrifices, and taboos, constituted an integral part of social customs and became a primary focus of scholars' criticism[1].

Among the many scholars of the Han Dynasty, Wang Chong 王充 (27–97) served as a local official for many years, during which he was exposed to beliefs in ghosts and deities in his daily life. His treatise *Lunheng* 论衡 (*Discourses Weighed in the Balance*) devotes significant space to analysis and criticism of beliefs in ghosts and deities, which is unprecedented regarding previous scholarly works. Current discussions on *Lunheng*'s depictions of these beliefs generally focus on Wang Chong's critique of the concept of ghosts and deities, as well as related customs. Initially, scholars regarded Wang Chong's critique through ideological and sociological lenses, viewing it as an important representative of "atheism"

and "materialism" (Hou et al. [1957] 1980; Satō 1981; Zhu 1989), and assigned his critique the significance of "class struggle" (Hou et al. [1957] 1980; Zhu 1989). Subsequent research mostly supplemented and refined the ideological interpretations, with some scholars questioning the "atheism" label (Deng 1997; F. Li 2014), others emphasizing Wang Chong's scientific and rational spirit (Tian 1981; W. Li 2000; Zhou 2015b), and some studying his logical methods to demonstrate the nonexistence of ghosts (Y. Xu 2021). In terms of the sociological aspect, however, while interpreting Wang Chong's critique through the perspective of "class struggle" became outdated, scholars rarely proposed new insights. Only a handful of scholars pointed out the relationship between Wang Chong's critique of customs and secular culture (Gong 2005; Zhang and Huang 2008). To date, there has been no research that has both conducted an overall analysis of Wang Chong's criticism of the belief in ghosts and gods and related customs from the perspective of sociocultural history and connected it with broader intellectual groups and discourses during the Han Dynasty.

Diverging from previous research, this article focuses on Wang Chong's critique of customs related to the belief in ghosts and deities, situating his criticism within the intellectual tradition advocating for social reform and custom changes. By comparing the diverse perceptions on customs and beliefs among the Han scholars, it explores the prevailing attitudes of scholars and the distinct personality of Wang Chong's thought, as well as how these varied stances intertwined with different life experiences, inherited academic traditions, and historical background. It aims to investigate the challenges faced by ancient Chinese scholars in navigating the intersections between rationality, morality, and religion and delve deeper into the characteristics of the beliefs of the Han Dynasty, as well as ancient China.

Furthermore, it is evident that the concern about ghosts and deities transcends cultural boundaries and is not solely a Chinese issue. In civilizations spanning Egypt, Mesopotamia, Greece, Rome, and beyond, beliefs and customs based on ghosts and deities constitute significant components of their respective cultures and societies. It can even be said that the history of human religion, to a certain extent, is a protracted struggle between humans and supernatural forces such as ghosts and deities. The skepticism and criticism surrounding these forces are also an indispensable part of the evolution of religion. As such, a thorough understanding of Wang Chong's perspectives not only sheds light on ancient Chinese religion but also serves as a pivotal case study in capturing the broader landscape of human religion in general.

## 2. The Concept of Ghosts and Deities

The core of the belief in ghosts and deities was the concept of ghosts and deities itself. Wang Chong's criticism of the belief in ghosts and deities also began with this concept.

In the Modern Chinese Dictionary, the first meaning of "ghost" refers to the spirit that exists after a person dies and leaves their physical body (Hanyu da zidian bianji weiyuanhui 1986, p. 4427). Similarly, the first meaning of "deity" or "god" refers to the creator and master of all things in the universe, as well as all spirits (Hanyu da zidian bianji weiyuanhui 1986, p. 2392). In early China, however, the word "ghost" (*gui* 鬼) referred not only to the soul of the deceased but also to the spirit of deities and even other non-human spirits (Poo 2022, p. 25). There are many relevant examples from ancient literature, such as *Mozi* 墨子, which states, "The ghosts of ancient and modern times are the same. There are the ghosts of Heaven, there are the ghosts and spirits of the mountains and rivers, and there are also the ghosts of people who have died [古之今之为鬼，非他也，有天鬼，亦有山水鬼神者，亦有人死而为鬼者]" (Johnston 2010, p. 303). In "Jiuge" 九歌 (Nine Songs) of Qu Yuan (BC340–BC278), the "mountain ghost" (*shangui* 山鬼) refers to the mountain deity. In *Rishu* 日书 (*the Day Book*) of the Qin Bamboo Slips from Shuihudi, "ghost" refers to harmful spirits with various origins. Therefore, some scholars believed that the original meaning of "ghost" was a general term that referred to the spirits or souls of humans, deities, and even animals. Evidently, both "ghost" and "deity" in early literature can refer to a variety of

spiritual beings, and the compound word "ghosts and deities" (*guishen* 鬼神) has always been used as a general term for various types of "spirits".

During the Spring and Autumn Period and the Warring States Period, there had already been a significant gap between the intellectual view on ghosts and deities and the popular view. The former viewed ghosts and deities as abstract existences that could not be sensed by human perceptions. As Confucius stated in *Liji* 礼记 (*The Book of Rites*): "How abundant and rich are the powers possessed and exercised by Spiritual Beings! We look for them, but do not see them; we listen for, but do not hear them; they enter into all things, and nothing is without them. They cause all under Heaven to fast and purify themselves, and to array themselves in their richest dresses in order to attend to their sacrifices. Then, like overflowing water, they seem to be over the heads, and on the left and right [鬼神之为德，其盛矣乎！视之而弗见，听之而弗闻，体物而不可遗。使天下之人齐明盛服，以承祭祀，洋洋乎！如在其上，如在其左右]" (Legge 1967, pp. 307–8). The ordinary people believed that ghosts and deities were tangible and visible and even possessed human-like consciousness, capable of influencing people's fortunes and misfortunes. Therefore, men of letters, especially the Confucian scholars, mainly adopted an attitude of reserving judgment and keeping a respectful distance from ghosts and deities, while the populace held a specific worship and fear of ghosts and deities.

The divergence between the two types of views on ghosts and deities is prominently reflected in the understanding of human ghosts formed after death. Since Zichan 子产 (?–BC522) proposed the theory of "soul and spirit" (*hunpo* 魂魄), from the Warring States Period to the Qin and Han Dynasties, a considerable number of scholars—not only Confucianists but also Taoists and Legalists—gradually formed a systematic view on life and death, ghosts and deities, soul and spirit, body and essence 形神, yin and yang 阴阳, heaven and earth (Zhou 2015a, pp. 321–22). Roughly speaking, the soul, essence, and yang energy belonged to heaven, while the spirit, body, and yin energy belonged to the earth. The combination of the two formed a human being. After death, the spirit, body, and yin energy returned to the earth, while the soul, essence, and yang energy returned to heaven, which were called ghosts and deities. In this system, the life and death of a person were explained in a "naturalistic" way. As a type of ghost and deity, human ghosts were also abstract and subtle beings that could not be seen or heard. By contrast, those who held the popular view on specific ghosts and deities often believed that "the dead become ghosts, are conscious, and can harm men [人死为鬼，有知，能害人]" (Forke 1962a, p. 191). What Wang Chong criticized was this view on specific ghosts and deities, especially the belief that human ghosts were tangible and conscious beings.

Firstly, Wang Chong inherited the concept of abstract ghosts and deities, as well as the naturalistic explanation of life and death. In the chapter "On Death", he pointed out:

"Ghosts and deities are only designations for something diffuse and invisible. When a man dies, his spirit ascends to heaven, and his bones return to the earth, hence they are called ghosts and deities. Ghost means 'to return', while deity refers to something diffuse and shapeless. Some say that ghosts and deities are names derived from yin and yang. Yin energy opposes things and makes them return, hence its name *gui* (ghost); yang energy fosters and produces things, and therefore is called *shen* (deity). Deity means extension, endless repetition, and circular renewal. Humans are born with divine energy, and when they die, they return to divine energy. Yin and yang are called ghosts and deities, and so are humans when they die". (Forke 1962a, pp. 191–92, with modifications)

[鬼神，荒忽不见之名也。人死精神升天，骸骨归土，故谓之鬼【神】。鬼者，归也；神者，荒忽无形者也。或说：鬼神，阴阳之名也。阴气逆物而归，故谓之鬼；阳气导物而生，故谓之神。神者，伸也，申复无已，终而复始。人用神气生，其死复归神气。阴阳称鬼神，人死亦称鬼神。]

Wang Chong also used the concepts of body and essence, heaven and earth, yin and yang to explain life and death, and he also regarded the ghosts and deities that people

transformed into after death as vague and invisible existences. In *Lunheng*, the terms "ghosts and deities" or "deities" mostly refer to such elusive and invisible beings[2]. Except for the aforementioned citation, there are other chapters such as "On Exorcism", which states, "Deities are diffuse, vague, and incorporeal: entering and departing they need no aperture, whence their name of deities [神，荒忽无形，出入无门，故谓之神]" (Forke 1962a, p. 536, with modifications); "The Knowledge of Truth", which states, "Deities are obscure, diffuse, and formless entities [神者，眇茫恍惚无形之实]" (Forke 1962b, p. 292, with modifications); "On Dragons", which states, "That which amidst Heaven and Earth is vague and unsubstantial as the vapors of cold and heat, wind and rain, has the nature of a deity [天地之间，恍惚无形，寒暑风雨之气乃为神]" (Forke 1962a, p. 353, with modifications); and "On Thunder and Lightning", which states, "Deities are diffuse and incorporeal. Departing or coming in they need no aperture, nor have they any hold above or below. Therefore one calls them deities [神者，恍惚无形，出入无门，上下无垠，故谓之神]" (Forke 1962a, p. 292, with modifications), etc.

Since humans become elusive, invisible, and unfathomable ghosts and deities after death, people should not pay much attention to the afterlife related to those ghosts and deities. Wang Chong said, "The dead are hidden from our view, being dissolved and belonging to another sphere than the living, and it is almost impossible to have a clear conception of them [死人暗昧，与人殊途，其实荒忽，难得深知]" (Forke 1962b, p. 370). The entire *Lunheng* does not discuss the afterlife, which is a direct manifestation of this attitude.

Based on the naturalistic view of life and death, Wang Chong further argued that "the dead do not become visible ghosts, have no consciousness and cannot injure others [人死不为鬼，无知，不能害人]" (Forke 1962a, p. 191, with modifications). He attacked the three aspects of the dead being tangible, the dead being conscious, and the dead being able to harm people mainly by explaining the relationship between the body and the essence. Wang Chong pointed out that, after death, the human body decayed and the essence dissipated, so it was impossible to have a physical form again; even if the essence still existed, without a physical form to attach to, it was difficult for it to be conscious and harm people. Therefore, the belief that "humans become visible existences after death, have consciousness, and can harm people" was a fallacy.

The relationship between the body and the essence was an important aspect of the naturalistic view of life and death. Wang Chong repeatedly emphasized the dependence of essence on physical form, echoing Xunzi's statement that "the body is set and spirit arises [形具而神生]" (Hutton 2014, p. 176), and he was also influenced by Huan Tan 桓谭 (BC23–56), whose impact was particularly significant.

Huan Tan once compared the relationship between spirit and physical form to that of a flame and a candle: "The spirit resides in the physical form, just like a flame burns a candle……Without the candle, the flame cannot exist independently in the void, and it cannot burn the wick after it has been extinguished. The extinguished wick is like an elderly person whose teeth have fallen out, hair has turned white, muscles have withered, and the spirit cannot nourish them. When the spirit and physical form are exhausted, life ends, just like a candle and flame burning out together [精神居形体，犹火之然烛矣。……烛无，火亦不能独行于虚空，又不能后然其烛。烛，犹人之耆老，齿堕发白，肌肉枯腊，而精神弗为之能润泽，内外周遍，则气索而死，如火烛之俱尽矣]". (Huan 2009, p. 32) Wang Chong not only inherited this viewpoint of the spirit's dependence on the physical form but also adopted the metaphor of flame and candle, indicating the profound influence of Huan Tan.

Wang Chong believed that ghosts and deities were elusive and invisible as smoke and clouds; thus, any tangible or visible existence could not be considered a ghost or deity. Wang Chong pointed out that "if he possesses a body, he does not deserve the name of a deity [如有形，不得谓之神]" (Forke 1962a, p. 293, with modifications). He once denied that the Thunder God (*leigong* 雷公) was a deity because it had a form. After death, humans joined the ranks of ghosts and deities, becoming elusive and invisible beings. Hence, the tangible ghosts seen by the populace were not real ghosts and deities and had no connection with the deceased.

If the visible ghosts that people saw were not the "spirit of the dead", then what were they? Wang Chong believed that visible ghosts were actually a kind of *yao* 妖 (uncanny phenomena). In Wang Chong's view, the strange events recorded in the classics, such as Zhao Jianzi's dream of the palace of the heavenly emperor, Liu Bang's killing of the white snake, and Zhang Liang's encounter with the old man named Yellow Stone, were all manifestations of *yao*. *Yao* was divided into different types, such as the *yao* of speech represented by nursery rhymes, the *yao* of sound represented by the crying of dry bones at night, and the *yao* of humans represented by witches. Similarly, the ghosts seen by people were "*yao* resembling human forms".

Wang Chong pointed out that the nature of *yao* was energy—to be more precise, "the energy of the grand yang" 太阳之气. Yin energy was mainly for flesh and blood, while yang energy was mainly for spirit. Only with both yin and yang energy could the spirit, flesh, and blood be solid and exist for long. However, the energy of the grand yang was only yang energy alone, so it could only be an image and did not have a physical form, which was too vague and turbulent to last long. Therefore, the manifestations of *yao* that presented human images—that is, the "ghosts" that people saw—were just illusions, not real entities. Their appearance often indicated the future, which was essentially similar to auspicious and calamitous portents.

The concept of *yao* is not an original creation by Wang Chong. In the pre-Qin period, literature such as *Zuozhuan* 左传 (*Commentary on the "Spring and Autumn Annals"*) and *Xunzi* 荀子already referred to abnormal and eerie phenomena as *yao*. In the context of the interaction between heaven and man, 天人感应 in the Han Dynasty, omens such as "*yao* of clothing" 服妖 and "*yao* of chicken" 鸡妖 were repeatedly discussed by Liu Xiang 刘向 (BC77–BC6), Jing Fang 京房 (BC77–BC37), Xiahou Sheng 夏侯胜 (?), and others, who believed those omens corresponded to human affairs.

Wang Chong's uniqueness lay in that he borrowed the concepts of yin and yang, form and essence to attempt a naturalistic explanation of the essence of *yao*, classifying it as "the energy of the grand yang". The energy of the grand yang belonged to the natural energy; the tangible ghosts, as a kind of *yao*, were also essentially natural energy, and their appearance followed the law of nature. Under Wang Chong's interpretation, the subjective consciousness of tangible ghosts was deprived, and the "supernatural" became "natural", a phenomenon that could be understood and explained; the connection between tangible ghosts and the good or bad fortune of people was no longer controlled by the consciousness of ghosts but rather unfolded naturally in accordance with the ways of heaven and earth.

In summary, on the one hand, Wang Chong negated the view that "humans become tangible ghosts after death, have consciousness, and can harm living people", based on the view of abstract ghosts and deities and the related naturalistic view of life and death; on the other hand, he used the concept of *yao* to provide a naturalistic interpretation of the generally perceived tangible ghosts. The former directly affected Wang Chong's attitude towards customs such as funerals, sacrifice, and taboos at that time; the latter was more related to Wang Chong's discussion on the interaction between heaven and man.

## 3. Customs Related to Ghosts and Deities

After thoroughly arguing that "people do not become tangible ghosts after death, they are unconscious and cannot harm others", thereby negating the prevalent belief in concrete ghosts and deities of the time, Wang Chong further discussed the real-world customs influenced by the concept of ghosts and deities.

### 3.1. Funerals

Firstly, Wang Chong addressed funeral practices. He explicitly stated, "the chapters on Death and Ghosts shall induce the ordinary people to give their dead a simple burial. [《论死》《订鬼》，所以使俗薄丧葬也]" (Forke 1962a, p. 90, with modifications), and "Now I have written the essays on Death and on the False Reports about the Dead to show that the deceased have no consciousness, and cannot become tangible ghosts, hoping that,

as soon as my readers have grasped this, they will restrain the extravagance of the burials, and become more economical [今著《论死》及《死伪》之篇，明【人】死无知，不能为鬼，冀观览者将一晓解约葬，更为节俭]" (Forke 1962a, p. 90, with modifications). Evidently, advocating for simpler burials was one of Wang Chong's primary focuses in discussing life, death, ghosts, and deities. The practice of extravagant burials was prevalent in the Han Dynasty, and there is ample documentary and archaeological evidence to support this, as detailed by previous scholars. Here, we only take the record from *Lunheng* as an example:

> "Ordinary people commiserate the dead that in their graves they are so lonely, that their souls are so solitary and without companions, that their tombs and mounds are closed and devoid of grain and other things. Therefore they make dummies to serve the corpses in their coffins, and fill the latter with eatables, to gratify the spirits. This custom has become so inveterate, and has gone to such lengths, that very often people will ruin their families and use up all their property for the coffins of the dead. They even kill people to follow the deceased into their graves, and all this out of regard for the prejudices of the living…… Therefore, the public remains wavering and ignorant, and those who believe in a lucky and unlucky destiny, dread the dead, but do not fear justice; make much of the departed, and do not care for the living. They clear their house of everything for the sake of a funeral procession". (Forke 1962b, pp. 369–70, with modifications)

> [闵死独葬，魂孤无副，丘墓闭藏，谷物乏匮，故作偶人以侍尸枢，多藏食物以歆精魂。积浸流至，或破家尽业，以充死棺；杀人以殉葬，以快生意。……是以世俗轻愚信祸福者，畏死（鬼）不惧义，重死不顾生，竭财以事神，空家以送终。]

The practice of lavish burials had a significant impact on people's lives at the time, even leading to the ruin of families and businesses, which can also be verified in other works in the literature. Regarding the origin of the extravagant burials in the Han Dynasty, numerous factors were believed to be pertinent, such as an extravagant atmosphere, economic prosperity, filial piety, and the increasingly specific imagination of the afterlife (Han 1998, pp. 94–96). In Wang Chong's view, the last factor was fundamental. The basis for imagining the afterlife lay in the belief that people were conscious after death. On one hand, people hoped to create a comfortable living environment for the deceased by burying rich items to prevent them from suffering, as they felt sorry for those who were buried alone; on the other hand, they hoped to avoid disasters and seek blessings by satisfying the needs of the deceased. Some grave-quelling texts (*zhenmuwen* 镇墓文) in the Han Dynasty with wishes such as "increasing wealth and population" (增财益口), "benefiting descendants" (利子孙), and "having no disasters" (无有央咎), reflect this mentality. Therefore, to eliminate extravagant burials, one needed to start by clarifying that people were unconscious after death, so that people could understand the futility of extravagant burials. Wang Chong pointed out, "Unless the discussion on death be exhaustive, these extravagant customs are not stopped, and while they are going on, all sorts of things are required for burials. These expenses impoverish the people, who by their lavishness bring themselves into the greatest straits [论死不悉，则奢礼不绝，不绝则丧物索用。用索物丧，民贫耗之至，危亡之道也]" (Forke 1962b, p. 374). Clarifying that people were unconscious after death was, in Wang Chong's view, the most thorough method to eliminate extravagant burials, which he believed previous scholars such as Jia Yi 贾谊 (BC200–BC168) and Liu Xiang had failed to achieve.

Here, it is necessary to briefly review the previous advocacy of simple burials before Wang Chong. During the pre-Qin period, the primary supporter of simple burials was naturally the Mohist school. Its standpoint was that extravagant burials were not beneficial to the social needs of enriching the poor, balancing the majority and minority, stabilizing crises, and governing chaos. The Confucian School did not clarify its stance on extravagant or simple burials. Despite later generations viewing Confucius as advocating for simple burials, Confucianism focused on "expressing emotions" and "following rites", emphasizing both genuine emotions and conformity to etiquette. Burials based on these principles

were considered reasonable. Later, in *Lüshi Chunqiu* 吕氏春秋 (*The Spring and Autumn Annals of Lü*), simple burials were advocated from the perspective of preventing tomb robbery and not dishonoring the deceased. In the Han Dynasty, *Huainanzi* 淮南子 criticized extravagant burials for exceeding reality, wasting resources, and harming the people, which was a rebuttal to the upper class's abuse of the funeral system in the guise of "rites". During the reign of Emperor Wu (BC141-BC87), Yang Wangsun 杨王孙(?), a Huang-Lao philosopher, advocated for simple burials, stating "the deceased are unconscious" and criticizing lavish funerals as wasteful and harmful. In *Yantie lun* 盐铁论 (*Treatise on Salt and Iron*), the virtuous and learned criticized the practice of extravagant burials as a luxury phenomenon in society, which also reflected the demand for simple burials. Under Emperor Cheng (reigned from BC33–BC7), Liu Xiang petitioned against the construction of Changling and Yanling, basing his argument on the social harms of extravagant burials and supplementing it with moral and fortune-telling theories. He noted that extravagant burials often led to tomb robberies and the collapse of the regime.

In general, the above-mentioned advocates of frugal burials mainly presented two orientations, either focusing on the social harm of lavish burials that wasted resources and labor or attempting to destroy the inherent belief of extravagant funerals based on the naturalistic view of life and death. The former accounted for the majority, while the latter was only seen in the expressions of *Huainanzi* and Yang Wangsun. Wang Chong emphasized that "people do not become tangible ghosts after death, they are unconscious and cannot harm others", which naturally belonged to the second type. Poo Mu-chou believed that those who advocated for simple burial based on the concept of universe creation and the view of life and death were mostly related to Taoism (Poo 1990, p. 563). This is indeed the case from the perspective of *Huainanzi* and Yang Wangsun, and, as an erudite scholar, Wang Chong was also deeply influenced by Taoist thoughts[3]. At the same time, Wang Chong noticed that, although the Confucian School held a naturalistic view of life and death, it did not clarify the point that "people are unconscious after death". No solid evidence was provided to prove this, let alone for using it as a weapon to oppose extravagant burials. Wang Chong believed that what prevented Confucianists from clarifying the point of "the dead have no consciousness" was the consideration of moral education:

> "Confucius perfectly well understood the true condition of life and death, and his motive in not making a clear distinction is the same which appears from Lu Chia's words. If he had said that the dead are unconscious, sons and subjects might perhaps have violated their duties to their father and sovereign. Therefore they say that the ceremony of funeral sacrifices being abolished, the love of sons and subjects would decrease; if they had decreased, these persons would slight the dead and forget the deceased, and, under these circumstances, the cases of undutiful sons would multiply. Being afraid that he might open such a source of impiety, the Sage was reluctant to speak the truth about the unconsciousness of the dead". (Forke 1962b, p. 372)

> [孔子非不明死生之实，其意不分别者，亦陆贾之语指也。夫言死【人】无知，则臣子倍其君父。故曰："丧祭礼废，则臣子恩泊；臣子恩泊，则倍死亡先；倍死亡先，则不孝狱多。"圣人惧开不孝之源，故不明死【人】无知之实。]

Wang Chong's interpretation of Confucius's intentions is closely related to a dialogue between Zi Gong and Confucius regarding whether the dead have consciousness, which was recorded by Liu Xiang. In this conversation, Confucius said: "If I said that the dead have consciousness, I would fear that filial sons and devoted grandchildren might do violence to themselves and follow the deceased to the grave; but, if I said that the dead have no consciousness, I would fear that filial sons and grandsons would cast aside their forebears' remains without properly burying them. [吾欲言死者有知也，恐孝子顺孙妨生以送死也；欲言无知，恐不孝子孙弃不葬也]" (Henry 2021, p. 1107). In other words, Confucianists were well aware that the dead had no consciousness, but due to the emphasis on filial piety and on the funeral system stemming from filial respect, they feared that once it was clarified

that the dead had no consciousness, the rituals of mourning and sacrifice would be abandoned, and the gratitude of subjects and children would be difficult to maintain, leading to moral corruption. Therefore, they adopted a deliberately ambiguous attitude.

Moral education is indeed an important factor, especially since Wang Chong himself attached great importance to it. Wang Chong's advocacy of simple burial was also motivated by the social responsibility of Confucian scholars, hoping to establish the teaching of simple burial and financial savings. However, Wang Chong believed that it was entirely possible to promote moral education by taking care of parents during their lifetime, and lavish burials after death were not beneficial to filial piety, government compensation, and people's enlightenment. Wang Chong's fundamental approach was to act based on reality rather than rigidly adhering to formalities. This is reflected in his statement: "When a sage has established a law furthering progress, even if it be of no great consequence, it should not be neglected; but if something is not beneficial to the administration, it should not be made use of in spite of its grandeur [圣人立义，有益于化，虽小弗除；无补于政，虽大弗与]" (Forke 1962b, p. 373). Even though lavish burial practices might appear to emphasize filial piety, their actual value paled in comparison to the detriment they caused to the national economy and people's livelihood. Therefore, such practices were not worthy of advocacy.

### 3.2. Sacrifices

Another custom that Wang Chong focused on was sacrifice. In the Han Dynasty, the scope of sacrifices was wide, including official sacrifices to heaven, earth, mountains, rivers, sun, moon, stars, heroes, and sages, as well as ancestor worship practiced by all social classes and sacrifices to various deities and spirits by the populace[4]. Since the pre-Qin period, scholars had mostly considered the officially recognized gods and spirits as worthy of sacrifice, believing that sacrifices to other ghosts and deities were "not in the sacrificial code" and belonged to excessive sacrifices (*Yinsi* 淫祀), which deserved criticism. Although Wang Chong also noticed the existence of numerous "excessive sacrifices" at that time, what he mainly criticized was the mentality of people sacrificing for blessings. In this sense, there was not much difference between "excessive sacrifices" and "non-excessive sacrifices", so Wang Chong did not specifically distinguish between the two but only commented on the general mentality of the worshipers:

> "The world believes in sacrifices, imagining that he who sacrifices will surely be blessed, and he who does not will surely be cursed. Therefore, when people are taken ill, they first try to learn by divination, what evil influence is the cause. Having found out this, they prepare sacrifices, and, after these have been performed, their mind feels at ease, and the sickness ceases. With great obstinacy they believe this to be the effect of the sacrifices. They never desist from urging the necessity of making offerings, maintaining that the departed are conscious, and that ghosts and spirits eat and drink like so many guests invited to dinner. When these guests are pleased, they thank the host for his kindness". (Forke 1962a, p. 509, with modifications)

> [世信祭祀，以为祭祀者必有福，不祭祀者必有祸。是以病作卜祟，祟得修祀，祀毕意解，意解病已，执意以为祭祀之助，勉奉不绝。谓死人有知，鬼神饮食，犹相宾客，宾客悦喜，报主人恩矣。]

It can be seen from the quotation that people believed "those who offer sacrifices will be blessed" on the premise that ghosts and deities were conscious and that they needed to eat and drink to maintain basic functions just like humans. Sacrifices could satisfy the dietary needs of ghosts and deities, thus exchanging them for their favor, gaining blessings, and avoiding disasters. Wang Chong was dismissive of the belief that "ghosts and deities can bring about misfortunes and fortunes". He noted that it was right to perform sacrificial rites, but it was wrong to believe in them, and used "ghosts and gods can eat and drink" as the main breakthrough point to criticize.

For the human ghosts among the ghosts and deities to whom people offered sacrifices, Wang Chong denied their possibility of enjoying food and drink based on the aforementioned naturalistic ideas about life and death: "Now those to whom we present sacrifices are dead; the dead are devoid of consciousness and cannot eat or drink [今所祭死人，死人无知，不能饮食]". (Forke 1962a, p. 509, with modifications) As for the non-human deities and ghosts, Wang Chong cleverly compared grand entities like heaven and earth, mountains and rivers, wind and rain, and celestial bodies to the intricate parts of the human body: bones, blood vessels, skin, flesh, and more. Through a meticulous examination, he pointed out the absurdity of imagining these bodily components relishing food and drink, concluding that non-human spirits cannot enjoy sustenance either. Consequently, Wang Chong asserted that neither human nor non-human spirits could savor nourishment. The inability of ghosts and deities to appreciate food implied their lack of consciousness and supernatural abilities, thereby rendering them unable to influence individuals' fortunes or misfortunes. In this way, the possibility of sacrifice bringing blessings or avoiding disasters did not exist.

On the basis of proving that sacrifices could not bring about good or bad luck, Wang Chong further demonstrated that fortune or misfortune had nothing to do with activities such as sacrifices:

> "When *Yao* and *Shun* practised their virtue, the empire enjoyed perfect peace, the manifold calamities vanished, and, though the diseases were not driven out, the Spirit of Sickness did not make its appearance. When *Chieh* and *Chou* did their deeds, everything within the seas was thrown into confusion, all the misfortunes happened simultaneously, and although the diseases were expelled day by day, the Spirit of Sickness still came back. Declining ages have faith in ghosts, and the unintelligent will pray for happiness. When the *Chou* were going to ruin, the people believed in ghosts, and prepared sacrifices with the object of imploring happiness and the divine help. Narrow-minded rulers fell an easy prey to imposture, and took no heed of their own actions, but they accomplished nothing creditable, and their administration remained unsettled.

> All depends upon man, and not on ghosts, on their virtue, and not on sacrifices". (Forke 1962a, pp. 534–35)

> [行尧、舜之德，天下太平，百灾消灭，虽不逐疫，疫鬼不往；行桀、纣之行，海内扰乱，百祸并起，虽日逐疫，疫鬼犹来。衰世好信鬼，愚人好求福。周之季世，信鬼修祀，以求福助。愚主心惑，不顾自行，功犹不立，治犹不定。故在人不在鬼，在德不在祀。]

As can be seen from the citation, Wang Chong believed that fortune and misfortune were directly related to moral behavior and were not affected by sacrifices. Undoubtedly, this was the inheritance of the humanistic and rationalist thought of the intellectual elite since the Spring and Autumn Period (Chen 2009). Similar discussions in the Han Dynasty mostly came from Confucian scholars after the mid-Western Han Dynasty, such as Du Ye杜邺 (?–BC2), Liu Xiang, and Huan Tan. These Confucian scholars attempted to link fortunes and misfortunes to moral gains and losses, thereby downplaying the power of religious activities such as sacrifices and divination. Their central point was what Wang Chong said, "All depends upon man, and not on ghosts, on their virtue, and not on sacrifices".

Unlike his predecessors, although acknowledging the correlation between fortune and morality, Wang Chong, at the same time, believed that good and bad luck were predetermined by fate: "If by offerings, happiness could be obtained, or if misfortune could be removed by exorcism, kings might use up all the treasures of the world for the celebration of sacrifices to procrastinate the end of their reign, and old men and women of rich families might pray for the happiness to be gained by conjurations with the purpose of obtaining an age surpassing the usual span. Long and short life, wealth and honour of all the mortals are determined by fortune and destiny, and as for their actions, whether they prove successful or otherwise, there are times of prosperity and decline [如祭

祀可以得福，解除可以去凶，则王者可竭天下之财，以兴延期之祀；富家翁妪可求解除之福，以取逾世之寿。案天下人民，夭寿贵贱，皆有禄命；操行吉凶，皆有衰盛]" (Forke 1962a, p. 535). This contradicts the aforementioned moral causality, because if fortune and misfortune are predetermined, then moral cultivation loses its meaning. Wang Chong did not resolve this contradiction, which also reveals the diversity and complexity of his thoughts.

Since sacrifice did not bring about blessings, and blessings did not come from sacrifices, what was the real meaning of sacrifice? After extensively quoting documents such as *Liji*, Wang Chong summarized that there were two meanings of sacrifice; one was to repay the merit of all the deities, and the other was to honor the deceased, which motivated individuals to emulate the virtuous deeds of past sages and respect their ancestors, and consequently enhanced and refined social customs. In this way, sacrificial rituals served more as a human practice to cultivate moral sentiments, strengthen family ties (Yang 1961, pp. 48–53), and harmonize social relations, rather than a supernatural matter of seeking blessings and avoiding disasters. Even though the rituals formally required solemn and respectful offerings to ghosts and deities, it was merely a symbolic act focusing on expressing inner feelings, which did not imply that the spirits would truly enjoy the offerings.

Wang Chong's understanding of sacrifice was the inheritance of the Confucian view of sacrifice since the pre-Qin period. Yu Yingshi once pointed out that in the pre-Qin period and in later generations, only the Confucian School followed Confucius' teaching of "sacrificing to deities as if the deities were present" 祭神如神在 and developed a more rational view of sacrifice (Yu 2005, p. 89). As Xunzi said, "The sacrificial rites are the refined expression of remembrance and longing. They are the utmost in loyalty, trustworthiness, love, and respect. They are the fullest manifestation of ritual, proper regulation, good form, and proper appearance. If one is not a sage, then one will not be able to understand them. The sage clearly understands them. The well-bred man and the gentleman are at ease in carrying them out. The officials take them as things to be preserved. The common people take them as their set customs. The gentleman regards them as the way to be a proper human being. The common people regard them as serving the ghosts. [祭者，志意思慕之情也，忠信爱敬之至矣，礼节文貌之盛矣，苟非圣人，莫之能知也。圣人明知之，士君子安行之，官人以为守，百姓以成俗。其在君子，以为人道也；其在百姓，以为鬼事也]" (Hutton 2014, p. 216). The terms "the way to be a proper human being" and "serving the ghosts" accurately describe the different understandings of sacrifice between Confucian scholars and ordinary people and also reveal the different positions of these two groups towards religious behavior.

Wang Chong's understanding of sacrifice also shows that, although he rejected the association between good or bad luck and sacrifice, he did not refute the legitimacy of sacrifice's existence. Instead, he attempted to interpret it through the lens of etiquette and righteousness, which contrasts significantly with his attitude toward extravagant burial practices. This divergence could be attributed to several factors. Firstly, sacrifice had deep historical roots, tracing back to ancient times, whereas extravagant burials were a more recent phenomenon without substantial theoretical and practical support. Secondly, Wang Chong's views might have been influenced by pragmatic considerations. Although extravagant burials served an educational purpose in promoting filial piety, they often resulted in undue financial and labor burdens, outweighing any potential benefits. Conversely, despite sacrifice having led to certain unhealthy trends in society, it had not caused significant harm to the national economy and people's livelihood compared to extravagant burials. In this way, Wang Chong's recognition of sacrifices and denial of extravagant burials reflected a value orientation based on reality.

### 3.3. Taboos

In the "Criticisms on Noxious Influences" chapter of *Lunheng*, there is such a description: "It is a common belief that evil influences cause our diseases and our deaths, and that in case of continual calamities, penalties, ignominious execution, and derision there has been some offence. When in commencing a building, in moving our residence,



in sacrificing, mourning, burying, and other rites, in taking up office or marrying, no lucky day has been chosen, or an unpropitious year or month have not been avoided, one falls in with demons and meets spirits, which at that ominous time work disaster [世俗信祸祟，以为人之疾病死亡，及更患被罪，戮辱懽笑，皆有所犯。起功、移徙、祭祀、丧葬、行作、入官、嫁娶，不择吉日，不避岁、月，触鬼逢神，忌时相害]" ([Forke 1962a](#), p. 525). It shows that there were various taboos in every aspect of life at that time. People believed that violating these taboos would irritate ghosts and deities, leading to disasters. As Jin Ze stated, taboos belong to "negative behavioral norms" (Z. [Jin 1998](#), p. 19). If the main mentality of people offering sacrifices was to seek blessings, then the mentality of observing taboos was to avoid disasters.

When summarizing the characteristics of taboos, Jin Ze mentioned that taboos carry fatal and mysterious dangers, and any violation of these taboos would result in mandatory punishment. Some of these dangers and punishments can be traced back to certain deities, while others are inexplicable, but all are related to the supernatural world (Z. [Jin 1998](#), p. 21). In Wang Chong's writings, we can also see the distinction between these two types of taboos. For instance, the four prevalent taboos described in *Lunheng* include "avoiding expanding the house westward" (讳西益宅), "avoiding visiting ancestors' graves after being punished and becoming a convict" (讳被刑为徒不上丘墓), "avoiding having contact with a woman who is giving birth to a child" (讳妇人乳子), and "avoiding bringing up children born in the first or fifth lunar months" (讳举正月五月子). These taboos belonged to the "inexplicable" rules, and people of that time often remained silent about their origins. The taboo of "whoever sees a two-headed snake will die" as recorded in the case of Sun Shu'ao in the "Wrong Notions about Happiness" chapter, also belonged to this category. Wang Chong once raised questions about the taboo of "avoiding expanding the house westward":

> "The craftsmen and technicians in the various arts and professions, in explaining omens, specify the different cases. The house builders state that in erecting a house mischievous spirits may be met with, in removing one's residence care should be taken to avoid the spirits of the year and the months, in sacrificing, certain days may be encountered when bloodshed is to be shunned, and in burying one may fail against the odd and even days. In all these instances these prohibitions are given because of ghosts and spirits, and evil influences. Those who do not avoid them, fall sick and die. But as for expanding the house westward, what harm is there, that it is held to be inauspicious, and how does the subsequent calamity manifest itself?" ([Forke 1962b](#), p. 377, with modifications)

> [诸工技之家，说吉凶之占，皆有事状。宅家言治宅犯凶神，移徙言忌岁月，祭祀言触血忌，丧葬言犯刚柔，皆有鬼神凶恶之禁。人不忌避，有病死之祸。至于西益宅何害，而谓之不祥？不祥之祸，何以为败？]

This shows that people did not understand why "expanding the house westward" was considered harmful, which differs from the second type of taboo described by Wang Chong, namely the taboos related to "ghosts and spirits, and evil influences" mentioned by the "craftsmen and technicians". In these taboos, the dangers and punishments were attributed to certain ghosts and deities, including mischievous spirits, the spirits of the year and the months, etc. Therefore, the second type of taboo was more closely linked to beliefs in ghosts and deities and thus became the main target of Wang Chong's criticism.

The so-called "craftsmen and technicians" clearly fall under the category of "workers" among the four social classes, specifically referring to those engaged in numerology and divination within the handicraft industry[5]. The taboos mentioned by the craftsmen and technicians are concentrated in several chapters of *Lunheng.* Among them, "Criticisms on Certain Theories" discusses taboos related to residential houses, "Questions about the Year Star" talks about taboos about removing one's residence concerning the Taisui 太岁 (the Year Star), "False Charges against Time" addresses taboos concerning ground breaking and house erecting related to the Taisui and Yuejian 月建 (the direction of the top of

the handle of the Big Dipper), and "Slaudering of Days" enumerates taboos concerning the selection of auspicious days for various events including funerals, sacrifices, bathing, tailoring, learning calligraphy, and so forth. These taboos exhibited continuity and regularity, appearing more as part of certain "rules". For instance, the theory of drawing plans of houses (*tuzhaishu* 图宅术), stating "the doors of a house of a family with a *shang* surname should not face the south, and that the doors of a house belonging to a family with a *chih* surname should not be turned to the north [商家门不宜南向，徵家门不宜北向]" (Forke 1962b, p. 416), reflected the rule of the Five Elements (*wuxing* 五行). The method for the moving of one's residence (*qianxifa* 迁徙法), stating "to encounter T'ai-sui is unlucky, and that to turn one's back upon it likewise bodes evil [徙抵太岁，凶；负太岁，亦凶]" (Forke 1962b, p. 402), reflected the rule of directions. The calendar for burials (*zangli* 葬历), stating "avoid the Nine Emptinesses, Depressions of the Earth, and consider the odd and even days, and single and paired months [葬避九空、地臽，及日之刚柔，月之奇耦]" (Forke 1962b, p. 393, with modification), reflected the rule of the Heavenly Stems and Earthly Branches (*ganzhi* 干支). The theory of drawing plans of houses, the method for moving, and the calendar for burials summarized these rules. If taboos were part of the rules, then the reasons for their establishment were naturally consistent with the reasons for the rules' establishment. Therefore, as mentioned earlier, for such taboos, people could clearly understand that their establishments were based on "avoiding the spirits of the year and the months" "bloodshed to be shunned" or "failing against the odd and even days". They all represented different rules[6].

It is important to note the relationship between the taboos mentioned by the "craftsmen and technicians" and the belief in ghosts and deities. Although these taboos were part of the rules, people often visualized the abstract rules as specific ghosts and deities. Typical examples include the taboos on moving and construction mentioned in the chapters "Questions about the Year Star" and "False Charges Against Time". These taboos were originally based on the rules of the movement of Jupiter and the Big Dipper (*beidou* 北斗), but people visualized these rules as the deities of Taisui and Yuejian, believing that violating the taboos would result in punishment from the deities. Wang Chong also pointed out that there were many gods on the almanac calendar, indicating that there were other taboos, especially taboos on choosing auspicious days, that were related to ghosts and deities. For instance, as exemplified in the chapter "Death" in *Rishu* from Kongjiapo, people at that time believed that illnesses occurring on specific Earthly Branches days were not attributed to the inherent auspiciousness or inauspiciousness of the Heavenly Stems and Earthly Branches themselves but rather to the offense of the deity associated with that particular Earthly Branch (Hubeisheng wenwu kaogu yanjiusuo and Suizhoushi kaogudui 2006, p. 172). Some scholars argued that the reason the Five Elements, Heavenly Stems, and Earthly Branches were perceived as ghosts and deities was because of the complexity of numerological studies. For ordinary people, explaining taboos in terms of ghosts and deities was more accessible and comprehensible (Liu 2007, p. 58). Therefore, although the taboos mentioned by the "craftsmen and technicians" were generally part of the abstract rules, they were often entangled with ghosts and deities in specific manifestations. In Wang Chong's writings, people at that time thought these taboos "are given because of ghosts and spirits, and evil influences" and believed that violating taboos would "offend ghosts and deities". Ban Gu 班固 (32–92) described that, after the scholars from the Yin-Yang School became "craftsmen and technicians", they were "restricted by taboos, obsessed with trivial matters, abandoning human affairs and relying on ghosts and gods [牵于禁忌，泥于小数，舍人事而任鬼神]" (Ban 1962, p. 1735), which was also based on the visualization of rules as ghosts and deities.

Wang Chong's critique of taboos is similar to his criticism of sacrifices. He first established his argument based on classics, demonstrating in detail that taboos cannot affect fortune; and then he argued that good or bad fortune had nothing to do with taboos but was predetermined by fate. This naturally stemmed from his fatalism. After breaking the connection between taboos and good or bad luck, Wang Chong attempted to clarify that

taboos were based on "rites and righteousness", which were intended to promote virtuous behavior, and there would be no harm from ghosts and deities. For instance, in the context of choosing auspicious dates, there were taboos such as avoiding Bing Day for studying and refraining from holding music events on Zi or Mao Days. The common belief was tied to legends like Cang Jie's death on a Bing Day or the fall of the Yin and Xia dynasties on Zi and Mao Days. Contrary to popular interpretations based on supernatural beliefs, Wang Chong argued that these taboos stemmed from a deep reverence for departed kings, reflecting a sentiment of sorrow and respect that discouraged celebrations on these specific days. Therefore, he postulated that many other taboos were likely founded upon the principles of ritual propriety and righteousness.

In short, Wang Chong's understanding of taboos is consistent with his broader perspective on religious practices such as sacrifices; while the masses saw taboos as religious practices to seek personal benefits and avoid harm, Wang Chong interpreted taboos through the lens of "rites and righteousness", regarding them as the result of the saints "using the beliefs as a way of moral education" 神道设教. In this way, he emphasized the social function of taboos, perceiving them as alternative embodiments of "rites" with the ultimate goal of social education. Thus, despite criticizing the widespread belief that taboos could influence fortune, he did not reject the legitimacy of their existence.

## 4. The Divergence in Scholars' Criticisms on Customs

Criticism of belief in ghosts and deities was not uncommon during the Han Dynasty. Confucian scholars before Wang Chong such as Gu Yong 谷永 (BC70–BC10), Du Ye, Liu Xiang, Yang Xiong 扬雄 (BC53–18), and Huan Tan, most of whom served in the imperial court, criticized the belief in ghosts and deities, primarily targeting rulers, who they hoped would abandon superstitious practices, assume cultural responsibility, and foster a positive social atmosphere. By contrast, Wang Chong, as a scholar serving in the locale, criticized the belief in ghosts and deities, mainly targeting ordinary people. When discussing issues related to the concept of ghosts and deities, funerals, sacrifices, and taboos, Wang Chong frequently used terms such as "secular" (世俗 *shisu*), "worldly" (世 *shi*), and "vulgar" (俗 *su*) to describe the believers. For instance, as previously stated, in the chapter "On Death", he wrote, "The world believes that the dead become ghosts, possess consciousness, and can harm people"; in "Simplicity of Funerals", he stated, "Therefore, the public remains wavering and ignorant, and those who believe in a lucky and unlucky destiny, dread the dead, but do not fear justice; make much of the departed, and do not care for the living".

Compared to those in high standing at court, Wang Chong had easier access to ordinary people in his daily life, and he primarily directed his criticisms towards that group of people. As Gong Pengcheng once summarized, criticism of customs in the Han Dynasty not only focused on criticizing the decadent atmosphere of emperors and aristocrats but also on the unsavory aspects of lives of ordinary men. During the Western Han Dynasty, criticism of customs focused on the former, but later, there was increasing concern for the daily customs and issues of ordinary people (Gong 2005, p. 41). Xu Xingwu also mentioned that since the end of the Western Han Dynasty, the pure Confucian scholars no longer served as political contenders in the central government or astrologers interpreting the will of heaven. Instead, they turned their attention to the entire country and local society (X. Xu 2005, p. 147). The writings of Wang Chong, as well as the subsequent criticisms of customs by Wang Fu 王符 (85–163), Ying Shao 应劭 (?), and Zhong Changtong 仲长统 (180–220), all discussed various secular issues, reflecting a shift in the political and intellectual concerns of scholars during the Eastern Han Dynasty. Undoubtedly, the focus on secular issues was also closely linked to the proliferation of local scholars, within which Wang Chong and his successors were all exemplary figures.

Secondly, when discussing the belief in ghosts and deities, most Confucian scholars before Wang Chong focused primarily on its external manifestations, without a deep exploration into the core concepts of ghosts and deities, particularly whether they were "con-

scious" or not. In terms of funeral customs, many Confucian scholars adopted an ambiguous stance. For example, Liu Xiang, when opposing Emperor Cheng on the construction of Changling and Yanling, stated, "If we presume the dead are conscious, then excavating graves would bring great harm; if they are unconscious, then what purpose does it serve [以死者为有知，发人之墓，其害多矣；若其无知，又安用大]?" (Ban 1962, p. 1956) This statement merely outlined the consequences of both "conscious" and "unconscious" scenarios without expressing a clear stance or viewpoint. As can be seen, Confucian scholars who disapproved of extravagant burials usually focused their arguments on practical harms such as exhausting resources and avoided the discussion on the nature of the deceased. Then, regarding sacrifices, Confucian scholars typically distinguished between excessive and appropriate worship, criticizing the former while tacitly approving the latter. The underlying beliefs in ghosts and deities behind sacrificial practices were rarely discussed either. As C. K. Yang once pointed out, "Many of the antisuperstition statements that have been quoted to prove their rationalistic intellectualism were motivated more by their struggle against organized religious movements which threatened their status than by well-reasoned, genuine skepticism about supernatural matters" (Yang 1961, p. 272).

Wang Chong, however, intentionally scrutinized and criticized the core ideologies that served as the foundation for funeral practices and sacrificial rites. This examination was notably reflected in his naturalistic approach to comprehending life and death, his emphasis on the unconscious state of the deceased, and his denial of supernatural qualities attributed to ghosts and deities. His goal was to profoundly undermine the very essence of these traditional customs. In comparison to his Confucian predecessors, Wang Chong's method marked a substantial change, as he incorporated diverse philosophical viewpoints into his argument, especially Taoist thoughts. Wang Chong's extensive assimilation of the teachings of Taoism and other academic schools mirrored the prevalent trend from the transition between the Western Han and Eastern Han dynasties to pursue wide-ranging knowledge beyond the dominant Confucianism, in which the revival of Taoist philosophy was particularly noteworthy. The intellectual pursuit of erudite scholarship persisted throughout the Eastern Han Dynasty and ultimately laid the foundation for the naturalistic philosophy that emerged in the subsequent Wei and Jin periods.

Furthermore, when it comes to the relationship between ghosts and deities and fortune, most Confucian scholars embraced the humanistic and rationalist ideologies that dated back to the Spring and Autumn period and the Warring States period. As a result, they diminished the power of ghosts and deities while highlighting the significance of human agency. This was prominently reflected in their association of good and bad luck with moral gains and losses, while downplaying the role of religious activities related to ghosts and deities such as sacrifices, divination, and taboos. Consequently, they forged a moralized universe.

Wang Chong, on one hand, declared that "All depends upon man, and not on ghosts, on their virtue, and not on sacrifices", emphasizing the importance of morality. On the other hand, he attributed the power originally belonging to ghosts and deities to fate, considering fate the ultimate external force: "Fate holds sway over happiness and misfortune, being a spontaneous principle and a decree to meet with certain incidents. There is no alien force, and nothing else exercises an overwhelming influence or affects the final result [命，吉凶之主也，自然之道，适偶之数，非有他气旁物厌胜感动使之然也]" (Forke 1962b, p. 1). Here, Wang Chong's views form an interesting contrast with Mohist thought; Wang Chong did not believe in ghosts and deities or a personified Heaven but firmly believed in fate, while Mohist essays, such as "Against Fate", "Heaven's Intention", and "Percipient Ghosts", denied omnipotent fate and assigned one's life encounters to the will of Heaven and the power of ghosts. This reveals the diversity of intellectual discourse when dealing with the intersection of religion, morality, and rationality. According to Wang Chong, fate held sway over happiness and misfortune, overseeing all aspects of life. Fate was formed by Heaven naturally imparting *qi* (vital energy) to humans, which seemed to have a physical origin. However, once a person received this *qi* and fate, their lifespan, fortunes, so-

cial status, wealth, and poverty were already determined: "Every mortal receives his own destiny: already at the time of his conception, he obtains a lucky or an unlucky chance [凡人受命，在父母施气之时，已得吉凶矣]". ([Forke 1962a](), p. 139) Thereafter, neither human effort nor extra-human powers could alter the predetermined destiny. Given that fate was determined before a person's birth, could steer the course of life, and was unaffected by any factors, most modern scholars attributed Wang Chong's theory of fate to fatalism (C. [Jin 2006](), pp. 450–55; F. [Xu 2014](), pp. 574–77; [Shao 2009](), pp. 297–98).

Wang Chong's admiration for fate surpassed his admiration for human morality, which also could be regarded as a deviation from Confucian tradition. Such deviation was closely related to Wang Chong's personal experiences of being unable to advance his career within the Han bureaucracy despite his talents and abilities, making him attribute his lack of success to fate ([Ma 2023]()). However, fatalism and moralism are fundamentally incompatible. As fatalism refers to human effort being unable to influence good or bad luck, Wang Chong's stance eliminated the significance of moral cultivation. Thus, Wang Chong's simultaneous advocacy for both morality and the role of fate when criticizing customs constituted a contradiction in his thought.

Following Wang Chong, Wang Fu also demonstrated confusion regarding the origins of fortune and misfortune. Wang Fu believed that both moral cultivation and fate influenced one's luck. While behavior was self-determined, fate was heaven-ordained. Ghosts and deities could also affect good and bad luck, but only through prayers' moral cultivation would the spirits be appeased and reward one with good fortune. Despite highlighting morality's significance, Wang Fu saw supernatural beings as mediators between human behavior and its consequences, believing that "when ghosts and gods are appeased, good fortune will flourish [鬼神受享，福祚乃隆]" (F. [Wang 2014](), p. 301). He actually viewed ghosts, deities, fate (heaven), and morality as sources of good and bad luck. In fact, thinkers throughout Chinese history struggled to reconcile the relationship between fortune and heaven, fate, supernatural beings, and morality. This was evident in the cases of Wang Chong, Wang Fu, and earlier scholars like Mozi. But in a society, it was difficult to put forward an explanation of human affairs that was universally accepted. For example, people who did not believe in gods or spirits often attributed various phenomena in the human world to Heaven or fate. Yet, it was challenging to explain human affairs entirely through the concepts of a willful Heaven or an immutable fate. Therefore, once scholars began to think, they often found themselves in a situation where almost no one had the perfect ability of logical and coherent thinking, making their ideas inconsistent and hard to justify. This dilemma was common among ancient Chinese scholars.

Finally, Wang Chong did not suggest abolishing all social customs such as sacrifices and taboos. Instead, he gave them a ritualistic interpretation, eliminating supernatural beliefs and focusing instead on the meanings recorded in Confucian classics. He aimed to connect small traditions with large traditions, ultimately serving the purpose of social education. As mentioned earlier, this approach was, to a large extent, inherited from Xunzi, and Wang Chong drew extensively from Xunzi's ideas throughout the entire *Lunheng* ([Yue 2006](), pp. 154–55).

Both Xunzi and Wang Chong explained and affirmed particular customs, such as sacrifices, as the external manifestations of ritual propriety and moral principles. However, Wang Chong diverged from Xunzi in his views on social customs. Xunzi endorsed the traditional Confucian moralism of "good or bad fortune depends on oneself 吉凶由人", perceiving morality as the crucial element that exerted a profound influence on one's success or failure. For example, when discussing physiognomy, Xunzi pointed out that what determined good or bad fortune lay not in external physical appearance but in the goodness or evil of one's heart. Those with good hearts were gentlemen, while those with evil hearts were petty men. "Becoming a gentleman is called good fortune. Becoming a petty man is called ill fortune [君子之谓吉，小人之谓凶]" ([Hutton 2014](), p. 32). For Wang Chong, however, the emphasis on the potent force of fate frequently surpassed his admiration for human endeavors. Consequently, when he reinterpreted customs on the basis



of rituals and righteousness, highlighting their moral connotations and promoting virtuous behaviors, he was not as convincing as Xunzi, and the efficacy of his approach was correspondingly diminished.

Additionally, apart from affirming a few customs from the ritualistic perspective, Xunzi viewed them in a mostly negative light and was even reticent to elaborate extensively on them, as seen by the recurrent assertion in the chapter "Against Physiognomy", where physiognomizing people was something "the ancients would not embrace, and the learned did not discuss [古之人无有也，学者不道也]" (Hutton 2014, p. 32, with modifications). While Wang Chong also, to some extent, criticized customs related to ghosts and deities, he steadfastly believed in other customs, including physiognomy. Furthermore, his extensive discussion of customs and beliefs, and abundant portrayal of folklore phenomena, frequently overshadowed and thus obscured his advocacy of Confucian ethics.

Ying Shao was similar to Wang Chong in this regard. In the preface to *Fengsu tongyi* 风俗通义 *(The General Meaning of Customs and Customs)*, Ying Shao mentioned that the purpose of his writing was to "correct the excesses and fallacies of popular customs and explain them through the principles of righteousness [言通于流俗之过谬，而事该之于义理也]" (Ying 1981, p. 4), that is, to rectify customs based on Confucian principles and reconcile them with classics and sages. Judging from the content of *Fengsu tongyi*, Ying Shao indeed provided interpretations of various customs and taboos such as peach sticks, reed stalks, painting tigers on doors, and sacrificing dogs at the four gates of the city through the lens of ritual propriety. However, due to the heavy focus on customs in the monograph, similar to the poetic device of "advising through a hundred examples while subtly criticizing one" (劝百讽一), he had difficulty highlighting his promotion of Confucian thought. Therefore, the Qing scholar Wang Mingsheng 王鸣盛 (1722–1797) commented, "Ying Shao is a vulgar Confucian of the Han Dynasty; Fengsu Tong is a petty work [劭，汉俗儒也；风俗通，小说家也]", and believed that *Lunheng* and *Fengsu Tongyi* belonged to the same category of books—"both are compilations of trivial knowledge and misinterpretations [皆摭拾謏闻，郢书燕说也]" (M. Wang 2010, p. 396). It is ironic, then, that the critics of customs were later regarded as their collectors and advocates. Be that as it may, their works are of exceptional significance in showcasing the customs and ideological variety in Eastern Han society.

## 5. Conclusions

Deeply embedded in the religious and cultural fabric of society, the belief in ghosts and deities reflects humanity's perpetual quest for meaning, order, and welfare. In ancient China, as individuals often lacked control over the fortunes and misfortunes of their lives, they resorted to seeking assistance from external forces, such as ghosts and deities, hoping that by appeasing these supernatural entities they could attain personal benefits. Popular beliefs in ghosts and deities in the Han Dynasty were often related to current careers, marriage, childbearing, and illnesses, and thus had nothing to do with the afterlife, reflecting distinct secular characteristics, which persisted throughout later generations and were unique among other religions of the world.

Simultaneously, Han Dynasty scholars' discussions on beliefs in ghosts and deities were also secular and not directly related to the religious background upon which these beliefs were based. Yu Yingshi pointed out that when scholars from early China criticized or reformed popular beliefs, they tended to only prohibit those that were extremely harmful to people's lives (Yu 1987, p. 197). That is to say, their primary consideration focused on material or practical aspects rather than pursuing the truth and purity of religious theories. The chief concern of scholars was to elevate social welfare, enlighten the public, and establish a utopian political sphere characterized by virtuous customs, a goal shared by Wang Chong as well.

Wang Chong's critique of the concept of ghosts and deities, as well as the customs of lavish burials, sacrificial rites, and taboos in the Han Dynasty, fell under the category of customs criticism within the intellectual tradition of reformers. Compared with his predecessors, Wang Chong's uniqueness was not only manifested in his overwhelming attention

to customs and beliefs but also in his extensive assimilation and utilization of various theories, including Confucianism and Taoism, when conducting criticism on customs. He dismantled the connection between "ghosts and deities" and "misfortunes and fortunes" with naturalistic views on life and death and theories of fate, undermining the inherent theoretical support shared by customs such as elaborate funerals, sacrifices, and taboos. Meanwhile, he maintained profound respect for tradition and culture, attempting to reinterpret customs as ritualistic practices rooted in Confucian principles, thus reconciling popular beliefs with ethical norms. Wang Chong's approach influenced subsequent figures such as Ying Shao, Wang Fu, and Zhong Changtong. These Eastern Han scholars, like Wang Chong, were concerned with folk customs and beliefs. Their emphasis on customs played a significant role in the transformation of intellectual climate from the elite culture to the secular culture during the late Eastern Han Dynasty and the Wei and Jin periods.

The study of Wang Chong's thought highlights the complexity of ancient Chinese religion, which was characterized by the interactions and tensions between the supernatural and the natural, between the sacred and the secular. Meanwhile, Wang Chong's approach of analyzing and criticizing the customs and beliefs offers valuable insights into how ancient Chinese scholars grappled with issues of belief, morality, and social custom, as well as how scholars can play a transformative role in promoting social welfare and cultural enlightenment in a wider scope today.

**Funding:** This research was funded by National Social Science Fund of China, Youth Project (国家社会科学基金青年项目), grant number 22CZS010.

**Institutional Review Board Statement:** Not applicable.

**Informed Consent Statement:** Not applicable.

**Data Availability Statement:** No new data were created or analyzed in this study.

**Conflicts of Interest:** The author declares no conflict of interest.

## Notes

[1]   Currently, there have been numerous research works regarding the relationship between men of letters and customs in the Han Dynasty, as well as the study of beliefs in ghosts and deities prevalent during that time. See Yu (1987), Poo (2007), J. Wang (2013, pp. 35–83), Hayashi (1974, pp. 223–306), and Ikeda (1981). However, most scholars who focused on customs adopted a perspective concerning customs overall and did not specifically address the customs related to beliefs in ghosts and deities. Conversely, researchers who studied beliefs in ghosts and deities tended to concentrate mainly on the beliefs themselves, with a notable lack of analysis on the attitudes held by scholars towards these beliefs.

[2]   In *Lunheng*, Wang Chong often used the term "ghosts and deities" to refer to elusive and invisible abstract beings, including human ghosts, and used the term "ghost" to refer to tangible, sentient, and potentially harmful beings in people's minds. As Poo Mu-chou once pointed out, the word ghost usually has a negative meaning when it appears alone in texts or in conversations, but when it appears together with the word god, the meaning is generally neutral. This can be somewhat confirmed in *Lunheng*. See Poo (2022, p. 2).

[3]   Wang Chong stated in the chapter "Simplicity of Funerals" that "Confucianists…… believe that the dead have no consciousness and cannot become ghosts", attributing the naturalistic interpretation of "the dead have no consciousness" to the Confucian School. The entire article only focuses on the pros and cons of Confucian and Mohist burial concepts, without mentioning the Taoist School. The reason may be that, as mentioned earlier, the naturalistic view of life and death had been shared by Taoists, Confucianists, and scholars from other schools of thought since the pre-Qin period. Many scholars in the Han Dynasty focused on integrating the strengths of various schools of thought. Even from the simple burial expositions of *Huainanzi* and Yang Wangsun, the integration of multiple schools of thought can be seen. Wang Chong's belief that the concept of "the dead have no consciousness" belonged to the Confucian School may be based on his overall Confucian standpoint.

[4]   It should be noted that there were many overlaps between official sacrifices and popular sacrifices in the Han Dynasty, from the identity of the worshippers to the objects of worship. The way to distinguish popular sacrifices from official sacrifices was as follows: (1) Those that were not in the official system were popular sacrifices. (2) The purposes of popular sacrifices and official sacrifices were different. The former pursued one's own happiness, while the latter focused on the welfare of the country. The Confucian scholars' understanding of sacrifices was not the same as the two above. Wang Chong's views can be seen as representative of the opinions of scholars.

5   The "House Moving School" 迁徙之家, "Five Elements School" 五行之家, and "Five Tones School" 五音之家 mentioned in *Lun-heng* also belong to the category of "craftsmen and technicians". Chen Pan identified the "craftsmen and technicians" as alchemists. See Chen (2010, pp. 189–90). While alchemists were skilled in both numerology and alchemy, with the latter (especially the pursuit of immortality) being the main focus, the craftsmen and technicians mentioned by Wang Chong were primarily engaged in numerology and rarely involved themselves in the pursuit of immortality through alchemy. Thus, there is a slight difference between the two.

6   Some taboos may have initially formed due to certain events, such as Wang Chong's example of "those who study calligraphy avoid the day of Bing" based on the death of Cang Jie on Bing Day, and "rituals avoid playing music on Zi and Mao Days" based on the demise of the Yin and Xia dynasties on the Zi and Mao Days. Similarly, the Qin Bamboo Slips from Shuihudi states, "Tianbo died on the day of Yi Si, Du died on the day of Yi You, Rain Master died on Xin Wei, and Tian Daren died on Gui Hai [田亳主以乙巳死,杜主以乙酉死，雨市（师）以辛未死，田大人以癸亥死]". See Shuihudi Qin mu zhujian zhengli xiaozu (1990, p. 368). Although specific taboos were not explicitly stated, it can be inferred that the formation of taboos was based on the deaths of Tianbo, Du, the Rain Master, and Tian Daren. However, most taboos ultimately became integrated into the framework constructed by various rules and existed as a part of those rules.

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
