# Peer review of "Exploring the Rational and Supernatural: Wang Chong’s Critical Analysis of Ghosts and Deities in Han Dynasty Customs"

_religions, doi:10.3390/rel15091094_

Round 1

Reviewer 1 Report

Comments and Suggestions for Authors

This paper is well structured and fluent. It centers on the Lunheng and systematically discusses the criticism of ghosts and spirits and associated customs by Wang Chong, the author of this book. The article does achieve the author’s purpose and has some academic value. The article does not have too many detail problems need to be corrected, but there are several fundamental problems that affect the judgment of its the academic value:

1. Hundreds of studies on Wang Chong and the Lunheng have been accumulated. Although this paper briefly mentions the shortcomings of the previous studies, it actually fails to systematize them. This results in the readers being unable to determine whether this paper is of sufficient value in advancing academic progress.

2. Following the previous problem, the article ignores a number of studies on related issues and fails to make a communication with these studies. Such as:

邓红:《试析王充的鬼神妖论》,《中国哲学史》1997年第3期;

徐英瑾:《王充的<论衡>是一部自相矛盾的哲学文本吗?》,《社会科学》2021年第12期;

李富祥:《王充<论衡>的命理学思想新探》,浙江师范大学,硕士论文,2014年;等。

3. The topic of this article appears to be a little outdated. Although the author provided a relatively good analysis and argumentation, there are already quite a number of established studies on this topic already. In other words, this article has made relatively limited progress in terms of academic innovation.

Overall, I believe this article would make a very good monograph chapter, but would be somewhat weak as a journal paper emphasizing innovation.

If this article was to be published, it would probably need to be supplemented with additional relevant researches and try to discover more illuminating new insights.

Reviewer 2 Report

Comments and Suggestions for Authors

1)     The author’s sources tend to be somewhat thin at strategic points.  For example, although Poo Muchou’s writings on ghosts and religious life is a first rate work, the author is somewhat too dependent on Poo and could improve the merit of the essay by broadening the sources, even if some (many) tend to agree or at least not contradict Poo’s work.  In working with Xunzi, the author would benefit from the use of Hutton’s translation of Xunzi rather than using Knoblock.

2)     Aside from the need for some more extensive sourcing, there are two areas of discussion that would improve the paper considerably.  First, some clarification about the difference between fate as Mozi understands it and as Wang thinks of it (lines 713-716 as a place to begin). Wang naturalizes ming and ties it to the yin/yang processes of qi.  It can certainly be argued that it is physics, not metaphysics that drives Wang’s account and this approach would be more consistent with the author’s general argument also.  This also means to speak of 'fatalism' as an ontology in the Western sense to describe Wang's view is arguable, if not simply false. Second, the author passes up several important opportunities to explain Xunzi’s influence on Wang (e.g., lines 492-503; 634-642; 738).  Some addressing of the following questions would be important.  What do we know about how much Wang was influenced by Xunzi?  Is there any substantial difference at key points in Wang’s view and that of Xunzi? If so, what are they? 

3)     Ill-advised rendering of yao , in line 199 and resurfaces in 216-222 and 223-232.  To use “demon” or “demons” for yao suggests a view that Wang did not hold, as he actually doesn’t believe in demons.  A preferable translation of the way he uses the term (although I’m not suggesting that every thinker used in this way) might better capture what he thinks is going on in such situations: “eerie,” “uncanny,” “preternatural,” “weird”.

4)     The section beginning with line 522 on Taboos might profit from a reference to Wang’s discussion of Sun Shu An.

5)     Stylistic….line 49 should be “summary”; line 155, perhaps “became” could better be “become”.  Include Chinese terms in lines 651-52

Round 2

Reviewer 1 Report

Comments and Suggestions for Authors

The revised manuscript has been greatly improved. Although there is still a lack of research depth, its research ideas and academic value are clear enough. The current paper has basically corrected several shortcomings I mentioned earlier. It basically meet the publication requirements.